# Overexpression of *NRF1-742* or *NRF1-772* Reduces Arsenic-Induced Cytotoxicity and Apoptosis in Human HaCaT Keratinocytes

**DOI:** 10.3390/ijms21062014

**Published:** 2020-03-16

**Authors:** Shuai Wang, Hao Cheng, Linlin Wang, Rui Zhao, Dawei Guan

**Affiliations:** Department of Forensic Pathology, School of Forensic Medicine, China Medical University, No. 77 Puhe Road, Shenyang North New Area, Shenyang 110122, China; shuaiwang0303@163.com (S.W.); chengh0920@163.com (H.C.); wangll@cmu.edu.cn (L.W.)

**Keywords:** NRF1, arsenic, HaCaT cells, cytotoxicity, post-translational modification

## Abstract

Increasing evidence indicates that human exposure to inorganic arsenic causes cutaneous diseases and skin cancers. Nuclear factor erythroid 2-like 1 (NRF1) belongs to the cap “n” collar (CNC) basic-region leucine zipper (bZIP) transcription factor family and regulates antioxidant response element (ARE) genes. The human *NRF1* gene is transcribed into multiple isoforms, which contain 584, 616, 742, 761, or 772 amino acids. We previously demonstrated that the long isoforms of NRF1 (i.e., NRF1-742, NRF1-761 and NRF1-772) are involved in the protection of human keratinocytes from acute arsenic cytotoxicity by enhancing the cellular antioxidant response. The aim of the current study was to investigate the roles of NRF1-742 and NRF1-772 in the arsenic-induced antioxidant response and cytotoxicity. We found that overexpression of *NRF1-742* or *NRF1-772* in human HaCaT keratinocytes decreased susceptibility to arsenic-induced apoptosis and cytotoxicity. In addition, we characterized the different protein bands observed for NRF1-742 and NRF1-772 by western blotting. The posttranslational modifications and nuclear translocation of these isoforms differed and were partially affected by arsenic exposure. Antioxidant protein levels were increased in the *NRF1-742* and *NRF1-*772-overexpressing cell lines. The upregulation of antioxidant protein levels was partly due to the translation of nuclear factor erythroid 2-like 2 (NRF2) and its increased nuclear transport. Overall, overexpression of *NRF1-742* and *NRF1-772* protected HaCaT cells from arsenic-induced cytotoxicity, mainly through translational modifications and the promotion of antioxidant gene expression.

## 1. Introduction

Inorganic arsenic (iAs) is a nonmetallic element widely found in nature, which causes global environmental health problems. It can enter the body in a variety of ways (e.g., respiratory, skin absorption, oral) [1,2]. Drinking arsenic-contaminated water is the main route of arsenic exposure [3,4]. Arsenic can have profound effects on multiple organs of the body [5,6,7,8,9]. The skin is the most sensitive and the first manifestation of arsenic toxicity [10,11]. About 17% to 66% of individuals chronically exposed to iAs will develop cutaneous diseases, such as hyperpigmentation [12], hypopigmentation [13], Bowen’s disease [14], and arsenic keratosis [15]. Approximately 1% of exposed individuals will develop squamous cell carcinoma or other types of skin cancer [16]. The mechanism of iAs toxicity has not been fully defined. Increasing evidence suggests that oxidative stress occurs in response to iAs exposure and could be a significant factor in dermal arsenic toxicity [17,18].

NRF1 and NRF2 belong to the cap “n” collar (CNC) basic-region leucine zipper (bZIP) transcription factor family and represent key factors that regulate the adaptive antioxidant response to oxidative stress [19,20]. Both NRF1 and NRF2 heterodimerize with the small Maf proteins in the nucleus and bind to the antioxidant or electrophile response element (ARE or EpRE, respectively) located in the enhancers of promoters to activate the transcription of their target genes. However, the regulation of NRF1 and NRF2 within cells differs significantly between the two proteins. Under normal conditions, NRF2 is maintained in the cytoplasm by Kelch-like ECH-associated protein 1 (KEAP1), where it is continuously ubiquitinated and degraded by the proteasome. In contrast, NRF1 is produced and glycosylated in the endoplasmic reticulum (ER) [21]. When oxidative stress occurs, NRF1 is deglycosylated and then cleaved by limited proteolysis into its active forms [22]. At the same time, the Keap1-NRF2 interaction is disassociated, allowing NRF2 to travel to the nucleus and activate the transcription of its target genes [23,24]. NRF1 is also involved in cellular immune responses, proteasome and metabolic homeostasis, and cellular differentiation. *NRF1* deficiency hinders the suppression of *inductive nitric-oxide synthase* (*iNos*) expression by transforming growth factor-β (TGF-β) [25]. Proteasome activity is damaged in the brains and livers of mice with *NRF1* conditional knockout [26,27]. The induction of proteasome genes by MG132 is impaired in *NRF1-*deficient fibroblasts [28]. Moreover, loss of *NRF1* in pancreatic β-cells results in increased basal insulin release and decreased glucose-stimulated insulin secretion [29]. NRF1 also regulates *dentin sialophosphoprotein (Dspp)* genes that are essential for the formation of bone and tooth [30].

The human *NRF1* gene is transcribed into multiple alternatively spliced transcripts, leading to the generation of multiple protein isoforms containing 584, 616, 742, 761, or 772 amino acids (aa) and deglycosylated forms. Our previous studies demonstrated that the long isoforms of NRF1 (L-NRF1) are involved in the protection of human keratinocytes from acute iAs^3+^ cytotoxicity by enhancing the cellular antioxidant response [31]. In addition, NRF1, NRF2, and KEAP1 participate in the coordinated regulation of the adaptive cellular response to iAs^3+^-induced oxidative stress [32]. However, the functions of the different NRF1 isoforms in iAs^3+^-induced HaCaT cell cytotoxicity are still unclear. Therefore, we established *NRF1-742*-overexpressing (OE) and *NRF1-772*-OE HaCaT cell lines to explore the function of these two isoforms. Furthermore, the deglycosylation and nuclear translocation of these isoforms were evaluated under normal or iAs^3+^-induced conditions. We found that overexpression of *NRF1-742* and *NRF1-772* increased the resistance of HaCaT cells to iAs^3+^-induced cytotoxicity.

## 2. Results

### 2.1. Characterization of Human Endogenous NRF1-742 and NRF1-772 Proteins and Their Derivative Isoforms

A presumptive schematic diagram of human *NRF1-742* and *NRF1-772* mRNA is shown in Figure 1A. To identify the specific NRF1-742 and NRF1-772 protein bands and assess the function of these isoforms in acute iAs^3+^-induced human keratinocyte damage, *NRF1-742* and *NRF1-772* were overexpressed in HaCaT cells by lentiviral transduction. We previously reported that the long isoforms of NRF1 were activated by iAs^3+^ in HaCaT and MIN6 cells [31,33]. Under normal conditions, NRF1-742 protein bands were observed at 78, 110 to 120, and 140-kDa (Figure 1B). The NRF1-772 protein isoforms were represented by bands of 78 and 150-kDa (Figure 1B). After a 6 h treatment with iAs^3+^, the intensity of these bands increased (Figure 1B). In addition, 120 to 140-kDa protein bands appeared in response to iAs^3+^ treatment in the *Cont*-OE cells. Multiple proteoforms of 95 to 120-kDa were present in *NRF1-742*-OE cells. In the *NRF1-772*-OE cells, iAs^3+^ induced 95 to 110 and 130-kDa bands. Bands of 30 and 60-kDa were observed in the *NRF1-772*-OE cells; however, these two bands did not change with exposure to iAs^3+^.

We determined the levels of the long isoforms of NRF1 in the subcellular fractions after iAs^3+^ exposure. Consistent with our previous results, the 78-kDa band and iAs^3+^-induced 120 to 130-kDa isoforms of *Cont*-OE accumulated mainly in the nuclear fraction (Figure 1C). In *NRF1-742*-OE cells, the 78-kDa and 95 to 120-kDa isoforms were found primarily in the nucleus after iAs exposure. Meanwhile, the 110 to 120-kDa bands were found in the cytoplasm (Figure 1D). The NRF1-772 protein bands of 78, 95 to 110, 130, and 150-kDa were present in both the cytosolic and nuclear fractions in response to iAs^3+^ treatment. Under normal conditions, the 78 and 150-kDa bands were mainly in the nuclear fraction (Figure 1E). The results are summarized in Table 1.

### 2.2. Modification of NRF1 Protein Isoforms by Glycosylation and Deglycosylation

Previous studies have reported that NRF1 is sequestered in the ER and the translocation of the N terminal into the ER results in glycosylation [34]. To detect the type of N-linked glycosylation and the generation of the different human NRF1 isoforms, we investigated the effect of two deglycosylases and the different ER stressors on the migration of iAs^3+^-induced NRF1 isoforms using SDS-PAGE.

In the *Cont*-OE cells, protein bands at 120 to 130-kDa and 140-kDa were diminished by treatment with Endo H and PNGase F (Figure 2A**–**B). Correspondingly, the abundance of the 78-kDa isoform was enhanced. BFA treatment resulted in the accumulation of a 150-kDa, suggesting the hyperglycosylation of the 140-kDa band (Figure 2C). Treatment with TU caused the faster migration of the 120 to 130-kDa isoforms (Figure 2C). TG slightly increased the abundance of the 120 to 130-kDa bands but did not affect their migration on SDS-PAGE (Figure 2C). Together, these data suggested that the 120 to 130-kDa and 140-kDa bands represented glycosylated proteins. 

In the *NRF1-742*-OE cells, both Endo H and PNGase F treatment caused decreases in the bands for the 95 to 120-kDa and 140-kDa isoforms (Figure 2D,E) and a slight increase in the intensity of the 78-kDa band. BFA slowed the migration of the 140-kDa isoform. Upon blockage of N-glycosylation with TU, the 95 to 120-kDa isoforms migrated faster (Figure 2F). These results indicated that the 95 to 120-kDa and 140-kDa isoforms were glycosylated proteins.

In the *NRF1-772*-OE cells, the 95 to 110-kDa and 150-kDa bands were diminished after Endo H and PNGase F treatment, while the 78-kDa band increased (Figure 2G**,**H). Surprisingly, the iAs^3+^-induced 130-kDa isoform was increased by Endo H but decreased by PNGase F, suggesting that the type of N-linked glycosylation differed for this isoform (Figure 2G,H). Treatment with BFA resulted in the slower migration of the 130-kDa band, and the accumulation of 150-kDa isoform (Figure 2I). After TU treatment, the 130 and 150-kDa isoforms migrated faster (Figure 2I). These results demonstrated that the 95 to 110-kDa, 130-kDa, and 150-kDa proteins were glycosylated in the ER.

### 2.3. NRF1-742-*OE* and NRF1-772-*OE* Cells Are Resistant to Acute iAs^3+^-Induced Cell Damage

To investigate whether NRF1-742 and NRF1-772 protected cells against acute iAs^3+^-induced cytotoxicity, we evaluated the effect of iAs^3+^ treatment on the cell viability of HaCaT cells. As shown in Figure 3A, iAs^3+^ caused a dose-dependent decrease in HaCaT cell viability. Overexpression of *NRF1-742* or *NRF1-772* caused resistance to iAs^3+^-induced cytotoxicity. Furthermore, the levels of apoptosis induced by a high concentration of iAs^3+^ were substantially lower in *NRF1-742*-OE and *NRF1-772*-OE cells compared to that in *Cont*-OE cells (Figure 3B,C). Consistent with this observation, the levels of cleaved caspase-3 and poly ADP-ribose polymerase (PARP) were reduced in *NRF1-742*-OE and *NRF1-772*-OE cells following acute iAs^3+^ exposure (Figure 3D,E). These results confirmed that overexpression of *NRF1-742* and *NRF1-772* protected HaCaT cells from the toxic effects of acute iAs^3+^ exposure.

### 2.4. Crosstalk among NRF1, NRF2, and KEAP1

We previously reported the existence of crosstalk among NRF1, NRF2, and KEAP1 [32]. Silencing of the long isoforms of NRF1 in HaCaT cells did not affect iAs^3+^-induced NRF2 accumulation but reduced the levels of KEAP1 in the presence or absence of iAs^3+^ [32]. In the present work, the levels of NRF2 in both the whole cell lysates (Figure 4A) and nuclear fractions (Figure 4C) from *NRF1-742*-OE and *NRF1-772*-OE cells treated with iAs^3+^ were slightly enhanced compared to the levels observed in *Cont*-OE cells. The NRF2 mRNA levels in the *NRF1-742*-OE and *NRF1-772*-OE cells were also enhanced by iAs^3+^ (Figure 4E). In contrast, KEAP1 expression levels were unaffected by overexpression of the long isoforms of NRF1 (Figure 4F).

### 2.5. Expression of NRF1 Target Genes Involved in the Antioxidant Response, Proteasome Function, and Nucleotide Excision Repair

To study the role of NRF1-742 and NRF1-772 in the iAs^3+^-induced antioxidant response, we measured the levels of glutamate-cysteine ligase catalytic (GCLC), glutamate-cysteine ligase modifier (GCLM) and NAD(P)H quinone oxidoreductase 1 (NQO1) by western blot. Acute iAs^3+^ exposure induced antioxidant protein expression in the *Cont*-OE cells (Figure 5A). Because the basal levels of the antioxidant proteins were relatively high in the *NRF1-742*-OE and *NRF1-772*-OE cells, iAs^3+^ treatment did not significantly alter the levels of these proteins (Figure 5A). Compared with the *Cont*-OE cells, the levels of the three antioxidant proteins in the *NRF1-742*- and *NRF1-772*-overexpressing cells were higher under normal or iAs^3+^ treatment, except for GCLM following iAs^3+^ treatment of *NRF1-742*-OE cells (Figure 5A–D). Analysis of the mRNA expression levels of the antioxidant genes (i.e., *NQO1, GCLC*, and *GCLM*) demonstrated that *NQO1* was expressed at higher levels in both the *NRF1-742*-OE and *NRF1-772*-OE cells following exposure to iAs^3+^ (Figure 5G). In the absence of iAs^3+^, *NRF1-742* and *NRF1-772* overexpression slightly decreased *GCLC* expression (Figure 5E). After iAs^3+^ treatment, however, *GCLC* expression in the *NRF1-742*-OE cells was significantly increased. In contrast, there were no differences in the *GCLM* mRNA levels between the three cell lines in the presence or absence of iAs^3+^. The mRNA levels of antioxidant genes could also be influenced by the different isoforms of NRF1 and NRF2. In general, antioxidant gene expression appeared to be involved in the iAs^3+^-induced antioxidant response. 

Accumulating evidence suggests that proteasome homeostasis is essential for cell survival, and it is maintained through feedback mechanisms where NRF1 levels are regulated by the proteasome levels [35,36]. We found that *PSMC3* and *PSMC4* were increased in *NRF1-772*-OE cells compared to the *Cont*-OE cells under either normal or stimulated conditions (Figure 5H,I). Expression of these genes did not change in the *NRF1-742*-OE cells. In summary, *NRF1*-overexpressing cells were partially resistant to iAs^3+^ exposure through effects on proteasome homeostasis.

We previously demonstrated that a deficiency in *NRF1* suppressed the transcription of xeroderma pigmentosum C (XPC), a factor indispensable for initiating nucleotide excision repair (NER) in HaCaT cells [37]. Furthermore, iAs^3+^ inhibited NER and XPC expression [38]. Consistent with these findings, XPC transcription was enhanced in both the *NRF1-742*-OE and *NRF1-772*-OE cell lines compared to *Cont*-OE cells before or after iAs^3+^ exposure (Figure 5J). In addition, iAs^3+^ reduced *XPC* in all three HaCaT cell lines. These data suggested that NER might be involved in the protection against iAs^3+^ -induced cytotoxicity and damage observed in *NRF1-742*-OE and *NRF1-772*-OE cells.

## 3. Discussion

Epidemiological studies have demonstrated that chronic exposure to arsenic can induce cutaneous lesions and skin cancers [39,40]. NRF1 can be transcribed and translated into multiple protein isoforms, and the human and murine long isoforms play protective roles against iAs^3+^-induced cytotoxicity [31,33]. The present study characterized the endogenous human NRF1-742 and NRF1-772 proteins and their derivative isoforms. Acute iAs^3+^ exposure induced additional protein bands than were observed under normal conditions. Some of the isoforms were processed in the ER and transported into the nucleus after post-translational modifications. *NRF1-742*-OE and *NRF1-772*-OE cells were partially resistant to iAs^3+^-induced toxicity. This phenomenon could be related to increased NRF2 levels and its nuclear accumulation, which modulated antioxidant gene expression.

The human NRF1 gene is transcribed into various isoforms of varying lengths (e.g., 583, 584, 616, 742, 761, and 772 aa). Xiang and Wang previously reported that the 742 aa isoform of human NRF1 and the 772 aa of human TCF11 appear as two major protein bands at around 140-kDa and two minor bands of approximately 120-kDa by western blotting [35]. We previously demonstrated that knockdown of the long isoforms of NRF1 led to distinctly diminished protein bands at 120 to 140-kDa [31]. In the present study, *NRF1-742*-OE cells exhibited protein bands of 78, 140, and 110 to 120-kDa, which were enhanced by iAs^3+^ exposure. NRF1*-*772 isoforms were observed at 30, 60, 78, 130, and 150-kDa by western blotting. Although the 30 and 60-kDa bands did not change upon iAs^3+^ treatment, isoforms at 95 to 110-kDa were induced. The differences observed between the current study and the previous reports may be due to differences in the antibodies used for the detection of NRF1, the methods used to generate the cell lines, the type of cells, and the reagents used to stimulate the cells. 

NRF1 is located in the ER, where it is glycosylated to an inactive form. When stimulation occurs, NRF1 is exported to the cytoplasm, deglycosylated, and finally truncated by proteases to generate its transcriptionally active forms that enter the nucleus and bind ARE-dependent promoters [39,41,42]. Lipid synthesis, protein folding, and protein maturation take place in the ER. Therefore, the accumulation of unfolded proteins and excessive protein trafficking can trigger ER stress. It is well-known that post-translational modification of NRF1 is important for its maturation and can be altered by ER stressors, including TU, BFA and TG. TU is an inhibitor of N-linked protein glycosylation. BFA inhibits protein transport from the ER to the Golgi, whereas TG blocks calcium uptake in the ER by inhibiting sarcoplasmic/endoplasmic Ca^2+^-ATPase. BFA, TU, and TG are distinct ER stressors that act through different molecular mechanisms [43]. Consistent with data from our laboratory and others [31,35], TU and BFA treatment altered the migration of NRF1 isoforms in SDS-PAGE gels. Following deglycosylation, only the 78-kDa protein band increased in all three HaCaT cell lines. The rest of the isoforms were visibly decreased. The results confirmed that the bands, with the exception of the 78-kDa band, were glycosylated isoforms in the ER. Through cytosolic fractionation experiments, we further found that the degree of glycosylation had an effect on nuclear accumulation. The glycosylated 150-kDa isoform of *NRF1-772*-OE partially remained in the cytosol under normal conditions. In contrast, the deglycosylated 78-kDa isoform was predominantly transported to fraction. In response to iAs^3+^ exposure, the glycosylated 130 and 150-kDa isoforms were present at relatively higher levels in the cytoplasm; however, the less glycosylated isoforms of 95 to 110-kDa were mostly present in the nucleus. Therefore, the status of glycosylation had a strong impact on NRF1 accumulation in the nucleus.

PNGase F and Endo H remove carbohydrate residues from proteins by different mechanisms. However, most studies on NRF1 glycosylation have only examined the effects of a single deglycosylase. PNGase F removes all types of *N*-linked (Asn-linked) glycosylation (e.g., high mannose, hybrid, and bi, tri, and tetra-antennary). Thus, this enzyme can remove all *N*-linked carbohydrates regardless of their type. In contrast, Endo H removes only high mannose and some hybrid types of *N*-linked carbohydrates, indicating that the type of *N*-linked glycosylation can be more closely determined using Endo H. In the present study, we found that the level of the 130-kDa isoform of NRF1 in *NRF1-772*-OE cells decreased following PNGase F treatment. However, it increased following Endo H treatment and the level of the 150-kDa isoform decreased. Together, we first demonstrated that the iAs^3+^-induced 130-kDa isoform contained a unique type of *N*-linked glycosylation, which might lack high mannose and hybrid types of glycosylation. Similarly, the other isoforms possessed high mannose and some hybrid types of *N*-linked glycosylation.

NRF1 and NRF2 are members of the CNC-bZIP family that regulate the adaptive antioxidant response by binding to the ARE. NRF1 and NRF2 compete with each other in response to oxidative stress via binding to Mafs and the ARE [19,44]. Our previous studies demonstrated crosstalk between KEAP1, NRF1, and NRF2 [32]. Increased NRF2 expression was observed in the absence of the long isoforms of NRF1 in HaCaT and MIN6 cells [32,33]. Based on our previous results, we predicted that NRF2 expression would be decreased by overexpression of *NRF1-742* or *NRF1-772*. However, we found excessive NRF2 accumulation in both the whole cell lysates and nuclear fractions from *NRF1-742*-OE and *NRF1-772*-OE cells; in addition, the level of KEAP1 did not change. These results might be due to the fact that the long isoforms of NRF1 also contain NRF1-761, which could exert a negative effect on NRF2 and KEAP1 expression. 

The production of reactive oxygen species (ROS) is thought to be the primary mechanism underlying arsenic toxicity [45,46]. Both NRF1 and NRF2 are known to regulate the expression of ARE-dependent genes [44]. In the present study, we found that the antioxidant protein levels were increased by iAs^3+^ exposure. In NRF1 and NRF2-overexpressing cells, the isoforms activated the ARE-dependent genes. iAs^3+^ can inhibit NER with a corresponding reduction in the levels of XPC, a protein that is critical for DNA damage recognition. The effect of iAs^3+^ on NER can partially account for iAs^3+^-induced cytotoxicity [38]. Inhibition of NER can sensitize cells to cell death and apoptosis induced by UV or docosahexaenoic acid and acrolein [47,48]. In our study, *XPC* mRNA expression was induced by the overexpression of *NRF1-742* or *NRF1-772* under normal conditions or following iAs^3+^ exposure. The induced XPC expression could potentially explain the iAs^3+^ resistance observed in the two *NRF1-*overexpressing cell lines. The ubiquitin-proteasome system (UPS) controls the degradation of most cellular proteins. The UPS plays an indispensable role in various biological processes, such as DNA repair, protein quality control, and the cell cycle. The accumulation of NRF1 in the cytosol and its processing are regulated by the proteasome in a p97-dependent manner, allowing NRF1 to promote cell survival by maintaining proteasome homeostasis under stressful conditions [49,50]. We found that *PSMC3* and *PSMC4* were affected by iAs^3+^ treatment, and the differences between the three cell lines were small, suggesting the effect of the long NRF1 isoforms on proteasome was weak.

## 4. Materials and Methods

### 4.1. Cell Culture and Reagents

HaCaT cells were obtained from N.E. Fusening at the German Cancer Research Center (Heidelberg, Germany). HEK-293T cells were purchased from ATCC (Manassas, VA, USA). Dulbecco’s Modified Eagle’s Medium (DMEM) and penicillin/streptomycin were purchased from Life Technologies (Shanghai, China). Fetal bovine serum (FBS) was from ExCell (Shanghai, China), and phosphate-buffered saline (PBS, pH 7.4, Part No. 02-023-1ACS) was from Biological Industries (BioInd, Israel). All cell culture dishes, flasks and plates were obtained from JET (Guangzhou, China). Sodium arsenite was purchased from Sigma-Aldrich (Saint Louis, MO, USA). Tunicamycin (TU), thapsigargin (TG), and brefeldin (BFA) were obtained from Calbiochem (San Diego, CA, USA). 

HaCaT cells were cultured in high-glucose DMEM supplemented with 10% FBS and 100 μg/mL penicillin/streptomycin at 37 °C with 5% CO_2_.

### 4.2. Lentiviral-Based NRF1 Overexpression

The lentiviral expression vector encoding *NRF1-742* (pLV07-human NRF1-742) was generated by digestion of pDONR223-human NRF1-742 (Open Biosystems, Waltham, MA, USA) with Nhe I and Xba I and subcloning of the NRF1-742 into the Nhe I site of lentiviral vector pLV07 (Biosettia, San Diago, CA, USA). The lentiviral expression vector encoding *NRF1-772* was generated by overlap extension PCR and subsequent cloning into the pLV07 vector. Cells transduced with the negative control lentiviral vectors are referred to as control cells. 

Cell transfection and lentiviral recombination and transduction were conducted as previously described [51]. In brief, the lentiviral transfer vector DNA, psPAX2 packaging (Addgene, Watertown, MA, USA) and pMD2.G envelope plasmid DNA (Addgene, Watertown, MA) weremixed at a ratio of 3:2:1 respectively. A total of 30 μg of the mixture was diluted to a total volume of 0.5 mL with distilled H_2_O and 125 μL of 1 M CaCl_2_ (Sigma, MO, USA). Following intense vortexing, 0.5 mL 2 × BES-buffered saline (280 mM NaCl, 50 mM HEPES, 1.5 mM Na_2_HPO_4_, pH 6.95) was added and incubated at room temperature for 30 min. After incubation, the solution was mixed again by gentle vortexing, and then added dropwise onto HEK-293T cells cultured in 10 cm dishes with 8 mL of DMEM without FBS. A total of 1 mL FBS was added two hours later. Sixteen hours post-transfection, the medium was replaced with DMEM supplemented with 10% FBS and incubated for 36 h before the first collection of viral supernatants. A second collection was conducted after a further 24 h. The supernatants from the two harvests were combined and cleared by centrifugation at 1500 rpm for 5 min at 4 °C and was then passed through a 0.45 μm pore MILLEX GP filter with a PES membrane (Millipore, Burlington, MA, USA).

HaCaT cells were infected with the lentiviral preparations described above for 3 days, and then the infected cells were selected using blasticidin S (5 μg/mL) for additional 3–5 days. Approximately 70% of the cells stably overexpressed *NRF1-742* and *NRF1-772*. 

### 4.3. In Vitro Deglycosylation 

In vitro deglycosylation with Endo H (P0702S, New England Biolabs) or PNGase F (P0704S, New England Biolabs) was carried out as described previously [34]. Briefly, 20 μg glycoprotein was heated at 100 °C for 10 min, combined with 1 μL of Glycoprotein Denaturing Buffer (B1704, New England Biolabs). The denatured samples were incubated at 37 °C for 1 h with 1000 units of Endo H or 500 units of PNGase F in a total reaction volume of 20 μL, containing 2 μL glycoBuffer 3 (B1720, New England Biolabs), or 2 μL glycoBuffer 2 (B3704, New England Biolabs) and NP40 (B2704, New England Biolabs) respectively. Samples were analyzed by western blotting.

### 4.4. Western Blot Analysis

Preparation of whole cell and nuclear lysates and western blotting was performed as described previously [31,32]. Briefly, whole cell lysates were prepared using cold RIPA lysis buffer supplemented with 2% phosphatase inhibitor cocktail (P1092, Beyotime, Shanghai, China) and 1 mM phenylmethylsulfonyl fluoride (ST506, Beyotime, Shanghai, China). Nuclear and cytoplasmic lysates were prepared using the Nuclear and Cytoplasmic Protein Extraction Kit (P0013B and P0028, Beyotime, Shanghai, China). Protein concentration was determined using the BCA method (P0011, Beyotime, Shanghai, China) according to the manufacturer’s instructions. Protein (50 μg) was separated by 10% or 12% SDS-PAGE and transferred to PVDF membranes. After blocking for 2 h in 3% BSA (Absin, China), the membranes were incubated with primary antibodies overnight at 4 °C. The antibodies against cleaved caspase-3 (#9664) and cleaved PARP (#5625) were purchased from Cell Signaling Technology (Danvers, MA). The antibodies against NRF1 (17062-1-AP), NQO1 (11451-1-AP), GCLC (12601-1-AP), KEAP1 (10503-2-AP), GCLM (14241-1-AP), tubulin (10094-1-AP), lamin A/C (10298-1-AP), and β-actin (60008-1-Ig) were obtained from Proteintech (Wuhan, China). The antibody against NRF2 (ab137550) was obtained from Abcam (Cambridge, MA, USA). The membranes were incubated for 2 h at room temperature with peroxidase-conjugated goat anti-rabbit IgG (ZB-2301, ZSGB-BIO, Beijing, China) or anti-mouse IgG (ZB-2305, ZSGB-BIO) and visualized using an ECL detection kit (Millipore, Billerica, MA, USA). The densities of specific bands were quantified by Image J software.

### 4.5. Cell Viability

Cell viability was analyzed as previously described [52]. HaCaT cells (10,000/well) were seeded in 96-well plates. The next day, the medium was replaced with fresh medium containing the indicated concentrations of iAs^3+^ for 24 h. Cell viability was assessed using the Cell Titer One Solution Cell Proliferation Assay (G3580, Promega, Madison, WI, USA) according to the manufacturer’s protocol. The cells were incubated with the reagent (20 μL) for 1 h at 37 °C with 5% CO_2_. The absorbance was read at 490 nm with a microplate reader (Molecular Devices, Sunnyvale, CA, USA).

### 4.6. RT-qPCR

Total RNA was extracted using Trizol reagent (Life Technologies, Carlsbad, CA) according to the manufacturer’s instructions. Total RNA was quantified using the Nanodrop One (Thermo, Wilmington, DE). Reverse transcription (RT) was performed on equal amounts of total RNA using the Prime Script RT Reagent Kit with cDNA Eraser (Takara, Dalian, China). SYBR Premix Ex Taq (Takara, Dalian, China) was used for subsequent qPCR. The specific primer sequences are listed in Appendix A. Real-time qPCR was carried out using the LightCycler 480 II (Roche, Basel, Switzerland). The data were analyzed using the ^∆∆^CT comparative method, and *β-*actin was used as the control gene for normalization.

### 4.7. Apoptosis Analysis

HaCaT cells were seeded in six-well plates and cultured to approximately 80% confluence. The cells were treated with iAs^3+^ at the indicated concentrations for 18 h. Both floating and attached cells were harvested for analysis. Cells were washed three times with pre-cold PBS and stained using the FITC Annexin V Apoptosis Detection Kit (556547, BD Biosciences, San Jose, CA, USA) according to the manufacturer’s instructions. Briefly, the cells were resuspended in 100 μL binding buffer with 5 μL Annexin V-FITC and 5 μL propidium iodide for 15 min at room temperature in the dark. Binding buffer (400 μL) was added to the cells, which were immediately analyzed using Beckman Coulter Cytoflex (Beckman Coulter, Kraemer Boulevard Brea, CA, USA). A total of 10,000 cells were acquired, and the percentage of apoptotic cells was determined using FlowJo 7.6.1 software (FlowJo LLC, OR, USA).

### 4.8. Statistical Analysis

All statistical analyses were performed using GraphPad Prism 7.0 (GraphPad, San Diego, CA, USA). The data are presented as the mean ± standard deviation of at least three independent experiments. The Student’s *t*-test was used for comparisons between two groups. Two-way analysis of variance (ANOVA) followed by the Bonferroni post-hoc test was used for comparisons between multiple groups. *p* < 0.05 was considered statistically significant.

## 5. Conclusions

In conclusion, the current study suggests that overexpression of *NRF1-742* or *NRF1-772* could increase resistance to iAs^3+^-induced cytotoxicity in human HaCaT cells. The underlying mechanism appears to be related to enhanced expression of antioxidant genes, nucleotide excision repair, and proteasome homeostasis, which may act synergistically to protect the cells. However, the two NRF1 isoforms possess distinct mechanisms of activation and post-transcriptional modification. Therefore, more research is needed on the NRF1 isoforms to fully comprehend their role in the dermal cell response to acute and chronic arsenic exposure.

## Figures and Tables

**Figure 1 ijms-21-02014-f001:**
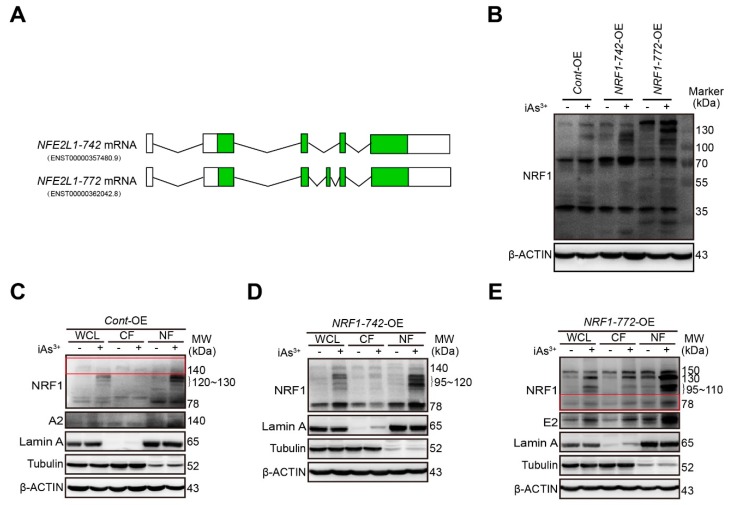
Detection of endogenous NRF1-742 and NRF1-772 proteins and their nuclear accumulation. (**A**) Schematic diagram of different isoforms of the human *NRF1* transcripts. Green and white open boxes represent the coding and untranslated regions, respectively. The solid black lines represent the introns. The sequences are from the National Center for Biotechnology (*www.ncbi.hlm.nih.gov*) and Ensemble Genome Browser (*www.ensemble.org*), updated as of November 2019. (**B**) NRF1 protein expression in HaCaT cells exposed to iAs^3+^ or vehicle (medium) for 6 h. Whole-cell lysates (WCL), cytosolic fractions (CF), and nuclear fractions (NF) from *Cont*-overexpressing (OE) (**C**), *NRF1-742*-OE (**D**), and *NRF1-772*-OE cells (**E**) following treatment with vehicle (medium) or 10 μM iAs^3+^ for 6 h. To better visualize the weak bands, the films with weak bands (red boxes) were cut for another longer exposure following the initial exposure of the whole membrane (labeled as A2, E2).

**Figure 2 ijms-21-02014-f002:**
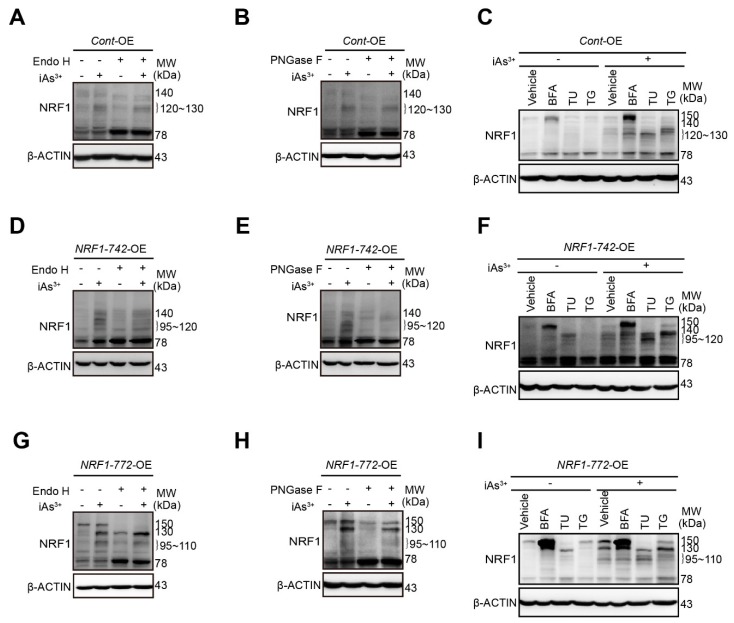
Effects of ER stressors and deglycosylase on NRF1 migration. Immunoblotting of NRF1 in whole cell lysates from *Cont*-OE (**A**–**C**), *NRF1-742*-OE (**D**–**F**), or *NRF1-772*-OE (**G**–**I**) cells 6 h post-treatment with the indicated agents. The doses of the agents were as follows: vehicle (0.5% DMSO), 10 μM iAs^3+^, 2 μg/mL Tunicamycin (TU), 1 μg/mL Brefeldin A (BFA), or 2 μM Thapsigargin (TG). (**A**,**D**,**G**) Samples were treated with Endo H (1000 U) at 37 °C for 1 h. (**B**,**E**,**H**) PNGase F-catalyzed deglycosylation (500 U) was performed for 1 h at 37 °C. Immunoblotting was performed with NRF1 and β-actin antibodies. Quantification of the NRF1 bands after deglycosylase treatment is shown in the Appendix A.

**Figure 3 ijms-21-02014-f003:**
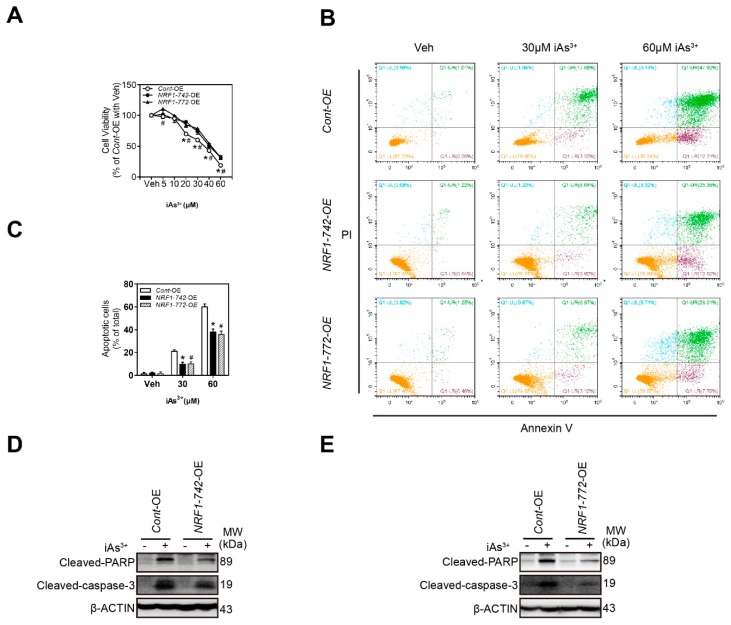
*NRF1-742*-OE and *NRF1-772*-OE HaCaT cells are resistant to the iAs^3+^-induced cytotoxicity. (**A**) Cell viability was assessed by the MTS assay 24 h of post- iAs^3+^ exposure (*n* = 6). The data are presented as the mean ± SD; * *p* < 0.05, *NRF1-742*-OE versus *Cont*-OE HaCaT cells; # *p* < 0.05, *NRF1-772*-OE cells versus *Cont*-OE HaCaT cells. (**B**) Representative flow cytometry images of Annexin Ⅴ and PI staining. (**C**) Quantitative analysis of apoptosis. Cells were treated with the indicated concentrations of iAs^3+^ for 18 h. The data are presented as the mean ± SD; * *p* < 0.05, *NRF1-742*-OE versus *Cont*-OE HaCaT cells; # *p* < 0.05, *NRF1-772*-OE cells versus *Cont*-OE HaCaT cells. (**D**,**E**) Cleaved PARP and caspase-3 immunoblotting of whole cell lysates from HaCaT cells treated with 30 μM iAs^3+^ for 6 h. β-actin served as the loading control.

**Figure 4 ijms-21-02014-f004:**
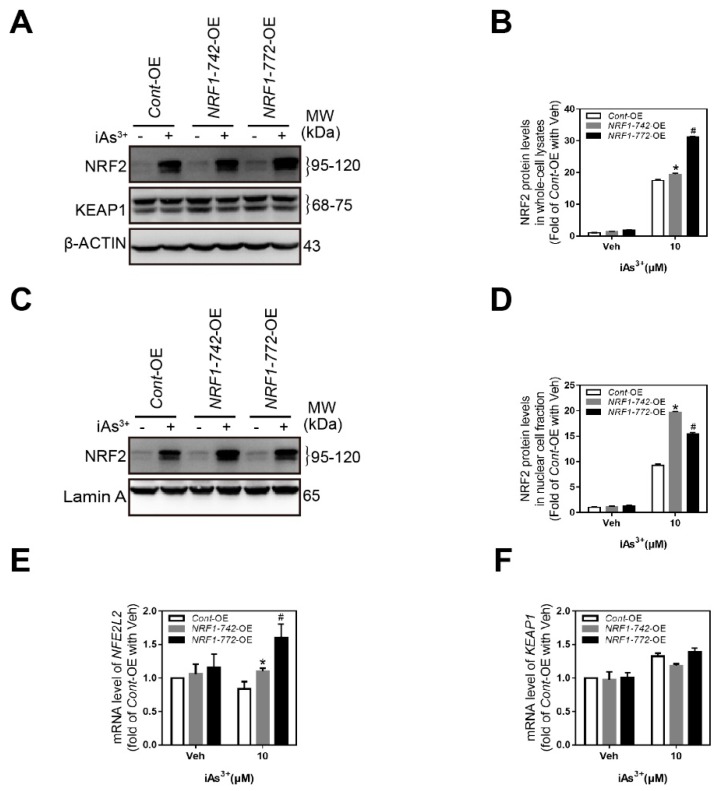
Crosstalk between NRF1, nuclear factor erythroid 2-like 2 (NRF2), and Kelch-like ECH- associated protein 1 (KEAP1). Cells were exposed to 10 μM iAs^3+^ or vehicle for 6 h. Immunoblotting of NRF2 in whole-cell lysates (**A**) and nuclear cell fractions (**C**). Quantification of NRF2 bands in whole-cell lysates (**B**) and the nuclear fractions (**D**). (E-F) RT-qPCR analysis of *NRF2* and *KEAP1* expression in response to acute iAs^3+^ exposure (*n* = 6). The data are presented as the mean ± SD; * *p* < 0.05, *NRF1-742*-OE versus *Cont*-OE HaCaT cells; # *p* < 0.05, *NRF1-772*-OE cells versus *Cont*-OE HaCaT cells.

**Figure 5 ijms-21-02014-f005:**
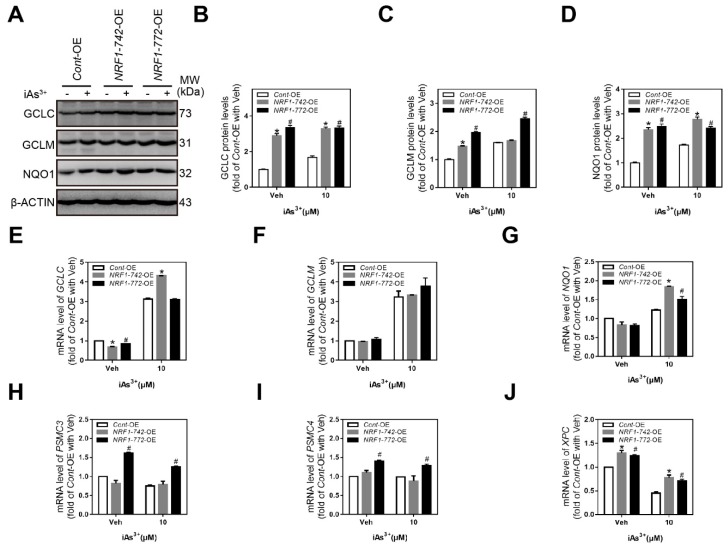
Expression of ARE-antioxidant genes, *PSMC3*, *PSMC4*, and *XPC*. (**A**) Antioxidant protein expression (GCLC, GCLM, and NQO1) was measured by immunoblotting after treatment with 10 μM iAs^3+^ or vehicle for 6 h. (**B**–**D**) Quantification of GCLC, GCLM, and NQO1 protein levels. (**E**–**J**) RT-qPCR analysis of the expression of antioxidant response element (ARE)-antioxidant genes (**E**–**G**), *PSMC3* (**H**), *PSMC4* (**I**), and *XPC* (**J**) following vehicle or iAs^3+^ treatment (*n* = 6). The data are presented as the mean ± SD; * *p* < 0.05, *NRF1-742*-OE versus *Cont*-OE HaCaT cells; # *p* < 0.05, *NRF1-772*-OE versus *Cont*-OE HaCaT cells.

**Table 1 ijms-21-02014-t001:** Subcellular localization of NRF1-742 and NRF1-772 proteins and their derivative bands under normal or iAs^3+^ treatment conditions.

Cell Type	Veh	iAs^3+^		
		WCL	NF	CF
*Cont*-OE	78, 140	78, 120–130,140	78,120–130,140	78, 140
*NRF1-742*-OE	78, 110–120, 140	78, 95–120, 140	78, 95–120,140	78, 110–120, 140
*NRF1-772*-OE	30, 60, 78,150	30, 60, 78,95–110, 130, 150	78, 95–110, 130, 150	78, 95–110, 130, 150

Whole-cell lysates (WCL); cytosolic fraction (CF); nuclear fraction (NF).

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
