# Peer review of "Overexpression of NRF1-742 or NRF1-772 Reduces Arsenic-Induced Cytotoxicity and Apoptosis in Human HaCaT Keratinocytes"

_ijms, 2020, doi:10.3390/ijms21062014_

Round 1
Reviewer 1 Report
In this study the authors assess the role of long isoforms of Nrf1 on the protection of HacaT cells against arsenic cytotoxicity.
There are some grammar and syntax errors that need to be corrected (please refer to the uploaded file). Why did the authors select these 2 isoforms? Why was not the isoform containing 761 aa investigated? How are the authors capable to distinguish close molecular weights on gels in Figures 1 and 2? The images do not seem clear to the reader for the observation of that subtle differences (e.g 120 and 130 kDa). Throughout the manuscript: Please replace "basal conditions" by "normal conditions" (please also refer to the uploaded file). Ln103: Please add the corresponding Figure (please also refer to the uploaded file). Ln104: Please add "Fig-1B" (please also refer to the uploaded file). Ln111: Do the authors mean green and white? Ln115, 116 and elsewhere: Please replace "protein" by "extract" or "fraction". Ln118-119: Please rephrase to make it clear to the reader. Ln123-124: This is not clearly shown in Fig 1D. Ln126: Not clearly shown in Fig-1E. Ln126: Please mention the corresponding Figure in the text (please also refer to the uploaded file). Ln131-138: Please move to the Discussion section. Ln144: How was this concentration of DMSO selected, given the different final concentrations of the different agents used? (please also refer to the uploaded file). Ln148: A downward shift of protein bands is observed around 140 kDa. Please discuss. Ln151 and 152: Please add (Fig-2C). Ln160: This is not clearly shown (please also refer to the uploaded file). Ln166-167: How is this shown? (please also refer to the uploaded file). Ln166-167: Please add corresponding figures (please also refer to the uploaded file). Figure 4B: Since these blots come from nuclear fractions, is the blot of β-actin correctly appearing here? Please check. Please 208 and 211: Please add corresponding figures (please also refer to the uploaded file). Please replace by more representative blots to what is shown in the densitometry graphs in Fig. 5B, C and D. Differences here seem to be very slight. Figure 5: Is there a different regulation at the transcriptional and translational levels? How can the authors explain differences in the induction at the mRNA level compared to the protein level? In addition, why expression of PSMC3, PSMC4, and XPC is not shown at the protein level? Ln420-422: ''The underlying mechanism appears to be related to enhanced expression of antioxidant genes, nucleotide excision repair, and proteasome homeostasis, which may act synergistically to protect the cells''. This is speculative. The authors should add some experiments in their study in which they will use specific for these particular pathways pharmacological inhibitors to confirm their implication in the observed phenomena.
Author Response
we provide the response to the reviewer's comments in a Word file as we uploaded.

Reviewer 2 Report
Dear Authors,
Please, find my suggestions in pdf file.

Author Response

(The authors gave the same response as above.)

Reviewer 3 Report
Reviewer Comments to Author(s)
This article investigated modification of NRF-1 long isoforms expressed in human keratinocytes. They also examined effects of overexpression of these isoforms on arsenic-induced cytotoxicity. Overall, this is a well-written manuscript. However, indicated immunoblotting images are something hard to understand. This might be improved such as overexpressing C-terminal tagged version of these isoforms and detect using an antibody against the tag. The acceptance of this article with current versions of figures is largely dependent on the judgement of the editor. Specific comments follow.
Page 1, line 36: Please indicate which isoforms the authors had investigated.
Page 2, line65: Insert a space between “stress” and “[19…”
Page 3, Figure 1A: Please indicate the target sequence of antigen for the used antibody.
Page 4, line 126: I cannot agree with the description “Under basal conditions, the….”.
Page 4, Table 1: Does the authors explain this table in the text?
Page 5, Figure 2A,B,D, Figure 4A,B, and others: Please indicate the quantitative data by using such as Image J.
Page 5, lines 155 and 162: We cannot predicate whether the 78 KDa form is deglycosylated here.
Page5, lines 159, 160 and others: It might be strange to describe “the OO-KDa isoforms migrated faster”.
Page 7, line 192: [32] should not be italic.
Author Response

(The authors gave the same response as above.)

Round 2
Reviewer 1 Report
This is the revised version of a previously submitted manuscript. I would like to thank the authors for their kindness to give me several explanations on my concerns, but I am afraid that the major issues of the work remain (please refer to my replies in the uploaded file) and are also shared by at least one of the other two Reviewers.
I believe the authors could try to resolve these issues and resubmit a more comprehensive and complete work.
I am sorry I could not be more positive on the current form of the study.
Author Response
We would like to reply to the comments in detail in the uploaded file.

Reviewer 3 Report
Reviewers comments to authors (Round 2)
Overall, the authors revised adequately according to reviewers’ comments. I hope they would investigate whether glycosylation and/or intracellular distribution of NRF1 affect the cellular tolerance to iAs treatment. I think this article might be acceptable for publication in Int. J. Mol. Sci.
Author Response

(The authors gave the same response as above.)
